# Barriers and Facilitators of Adherence to Nicotine Replacement Therapy: A Systematic Review and Analysis Using the Capability, Opportunity, Motivation, and Behaviour (COM-B) Model

**DOI:** 10.3390/ijerph17238895

**Published:** 2020-11-30

**Authors:** Amanual Getnet Mersha, Gillian Sandra Gould, Michelle Bovill, Parivash Eftekhari

**Affiliations:** 1School of Medicine and Public Health, The University of Newcastle, University Drive, Callaghan, Newcastle, NSW 2308, Australia; gillian.gould@newcastle.edu.au (G.S.G.); michelle.bovill@newcastle.edu.au (M.B.); parivash.eftekhari@newcastle.edu.au (P.E.); 2Hunter Medical Research Institute, Lot 1, Kookaburra Circuit, New Lambton Heights, Newcastle, NSW 2305, Australia

**Keywords:** adherence, COM-B model, factors, nicotine replacement therapy, smoking cessation

## Abstract

Background: Poor adherence to nicotine replacement therapy (NRT) is associated with low rates of smoking cessation. Hence, this study aims to identify and map patient-related factors associated with adherence to NRT using the capability, opportunity, motivation, and behaviour (COM-B) model. Methods: A systematic review was conducted by searching five databases (MEDLINE, Scopus, EMBASE, CINAHL, and PsycINFO) and grey literature on 30 August 2020. Data were extracted, thematically analysed, and mapped to the COM-B model. The Joanna Briggs Institute (JBI) critical appraisal tool was utilised to assess the quality of studies. Results: A total of 2929 citations were screened, and 26 articles with a total of 13,429 participants included. Thirty-one factors were identified and mapped to COM-B model: psychological capability (forgetfulness, education), physical capability (level of nicotine dependence, withdrawal symptoms), reflective motivation (perception about NRT and quitting), automatic motivation (alcohol use, stress, depression), physical opportunity (cost), and social opportunity (social support). The most prominent element associated with adherence was reflective motivation followed by physical capability and automatic motivation. Conclusions: Multiple personal, social, and environmental factors affect NRT adherence. Hence, it is recommended to implement a multifaceted behavioural intervention incorporating factors categorised under the COM-B model, which is the hub of the behaviour change wheel (BCW) to improve adherence and quitting.

## 1. Backgrounds

Tobacco smoking is one of the main public health concerns that the world has ever faced [1]. Since the adoption of the World Health Organisation (WHO) Framework Convention on Tobacco Control (FCTC) in 2003, tremendous efforts have been made to scale-up tobacco control [2,3,4]. Smoking cessation is one of the most important and cost-effective preventive health measures to reduce the risk of mortality and morbidity [5,6]. Smoking cessation is associated with substantial positive health outcomes, and evidence suggests that smoking cessation medications are offered in addition to behavioural therapy [7].

One factor that has been shown to have a direct effect on the success of smoking cessation treatment is adherence to smoking cessation medications [8]. A review conducted in 2020 by Mersha et al. showed that adherence to nicotine replacement therapy (NRT) doubles the success of smoking cessation (OR = 2.17, 95% CI, 1.34–3.51) [9]. A literature review conducted by Pacek et al. which included participants utilising any type of smoking cessation medications such as varenicline, bupropion, and NRT, classified factors associated with adherence to smoking cessation medications into preventable and non-preventable factors. The non-preventable factors include sociodemographic, medical comorbidities, genetic, and personality factors [10]. Male sex [11,12], older age [8,13], and greater educational status [14] were found to increase the level of adherence to smoking cessation medications. Low socioeconomic status and having depressive symptoms were found to reduce the level of adherence in most of the studies [15,16,17]. Preventable factors associated with adherence to smoking cessation medications include belief about the safety and efficacy of smoking cessation medications [18]. 

In the behaviour change wheel (BCW), which is broadly utilised to design and implement successful behavioural change interventions, the capability, opportunity, motivation, and behaviour (COM-B) model is at the hub of the wheel. The COM-B model suggests that behaviour is the result of an interaction between three components: capability, opportunity, and motivation. These components are further divided into six subcomponents: psychological capability, physical capability, social opportunity, physical opportunity, automatic motivation, and reflective motivation [19]. The components of the COM-B model are encircled by nine intervention functions (education, persuasion, incentivisation, coercion, training, restriction, environmental restructuring, modelling, enablement) and seven policy categories (communication, guidelines, fiscal, regulation, legislation, environmental/social planning, and service provision) in the BCW [20]. This representation in the BCW makes suggestion of possibly effective strategies easy and targeted to components of the COM-B model. 

Among the smoking cessation medications, NRT has the lowest half-life, especially nicotine gum and spray, with a maximum of two to three hours; whereas, the half-life of bupropion and varenicline is 22 and 24 h to ease adherence as compared to NRT [21,22]. Moreover, the rates of adherence were 63 and 74% for varenicline and bupropion, respectively [23,24]. Adherence to NRT is inconsistent between 26 to 61%, given the difficulty of the dosing schedule [9]. Although, a literature review conducted in 2018 aimed at classifying factors as preventable and nonpreventable [10], the current review utilised the COM-B model to understand factors and BCW to discuss and suggest interventions [20]. 

A detailed understanding of the barriers and facilitators of adherence to NRT is crucial for the development of comprehensive and effective interventions that can improve the success of smoking cessation. This systematic review aims to identify barriers and facilitators of adherence to NRT and to map the identified factors into the six sub-components of the COM-B model. As the COM-B model is the heart of the BCW, it will guide researchers and policymakers to develop targeted strategies that may improve adherence to NRT and smoking cessation [20]. 

## 2. Methods 

This systematic review was conducted according to the Preferred Reporting Items for Systematic Reviews and Meta-Analyses (PRISMA) guidelines [25]. The protocol was registered in PROSPERO (registration number CRD42020186621), available from https://www.crd.york.ac.uk/prospero/display_record.php?ID=CRD42020186621 [26]. This review followed five steps utilised by similar previously published reviews [27,28]: (1) review scope defined; (2) literature search conducted; (3) citations screened for inclusion; (4) data extracted and associated factors identified; and finally, (5) identified factors mapped to COM-B model. 

### 2.1. Selection Criteria

Population: Studies that enrolled individuals using NRT for smoking cessation were included in the systematic review. Studies were included without restriction for age and medical condition. 

Intervention: The interventions were the administration of any form of NRT including gum, transdermal patch, nasal spray, oral spray, lozenges, mini lozenges, or oral inhalator over various periods. Studies using multiple smoking cessation medications were included if they reported factors associated with only NRT users separately. 

Comparator: In studies with a control group, the control was either a placebo, active comparator group, or no intervention at all. 

Outcome: Studies that reported factors associated with adherence to NRT were included. 

Study: Both quantitative and qualitative study designs, such as cross-sectional surveys, case-control studies, longitudinal studies, qualitative studies, mixed-method studies, and clinical trials with full texts were included in the review. Commentary, abstracts, reviews, and editorial letters were excluded. There was no restriction regarding geographical locations, year of publication, and language of publication. 

### 2.2. Literature Searching and Citation Screening

Five electronic databases (MEDLINE, Scopus, EMBASE, CINAHL, and PsycINFO) were searched from the start of indexing to 30 August 2020. Furthermore, each reference list of included studies was checked and subject-based journals such as Nicotine and Tobacco Research and Journal of Addiction were searched. Grey literature such as the Centre for Disease Control and Prevention Smoking and Health Resource Library and National Institute for Health and Care Excellence were searched. Additionally, the first ten pages of free google search were revised for possible eligible articles. The selected databases and grey literature were searched using words and phrases displayed below. The search strategy was developed with an experienced librarian (Appendix A). Citations collected using Endnote reference management software version 9 (Clarivate, New York, NY, USA) and exported to Covidence software for screening [29]. Two reviewers (A.M. and D.T.) screened identified citations by using the above-mentioned inclusion/exclusion criteria.

### 2.3. Keywords

Smoking [MeSH Terms], Smoking cessation [MeSH Terms], cessation [all fields], smoke [all fields], cigarette [MeSH Terms], Quitting [all fields], Quitting Smoking [all fields]

“Medication Adherence” [MeSH Terms], Adherence [all fields], Discontinuation [all fields], Compliance [all fields], “Medication compliance” [MeSH Terms], Non-compliance [all fields], Non-adherence [all fields], “Treatment Compliance” [all fields], “Therapeutic Compliance” [all fields] 

“Nicotine replacement therapy” [all fields], NRT [all fields], “Nicotine patch” [MeSH Terms], Patch [all fields], “Nicotine gum” [MeSH Terms], “Nicotine inhaler” [all fields], Inhaler [all fields], Lozenge [all fields], “Nicotine spray” [all fields], Pharmacotherapies [all fields], “Drug therapies” [all fields], “Pharmacological therapy” [all fields], and “Medication treatment” [all fields].

### 2.4. Data Extraction

Data extraction for quantitative studies was performed by two reviewers (AM, PE) independently for each article. Data extraction from qualitative studies were performed using Nvivo 12 software (QSR International, Melbourne, Australia). Factors associated with adherence to NRT were identified, coded, and analysed using framework analysis. The data extraction template incorporated information on the identification of studies, methodological characteristics, and main findings regarding the barriers and facilitators of adherence to NRT. When there existed a disagreement between reviewers, it was resolved through discussion and mutual agreement.

### 2.5. Quality Assessment

The quality of each study was assessed using the Joanna Briggs Institute (JBI) critical appraisal tool [30]. Studies scoring ≥7 out of a maximum score of 10 or ≥70% if the maximum score was not 10 were considered high quality; studies scoring <4 or 40% were considered low-quality studies. Those studies scoring between 4 and 7 (40–70%) were considered to have medium quality. As illustrated in Table 1, most of the studies included in this systematic review were assessed to have high quality. All but three studies [31,32,33] scored over 70% on the Joanna Briggs Institute (JBI) critical appraisal tool [30]. The most common limitations among randomised control trials were failing to report whether those delivering treatment and outcome assessors were blind to treatment assignment [16,17,32,33,34,35,36,37,38,39,40,41]. Whereas, the most common limitations identified from observational studies were the absence of objective and standard criterion to measure the condition, which may increase the risk of bias of studies [8,31,42]. None of the qualitative studies included a statement about the influence of researchers’ culture or belief on the research [43,44,45,46]. (Appendix A).

### 2.6. Data Synthesis

Data were extracted and analysed using a framework analysis and factors associated with adherence to NRT were mapped into the six sub-components of the COM-B Model (psychological capability, physical capability, social opportunity, physical opportunity, automatic motivation, and reflective motivation) using a guideline set out by Michie and colleagues [19]. Two reviewers (A.M. and P.E.) independently mapped the factors and disagreements were resolved through discussion and mutual understanding. Factors under the six sub-components of the model are discussed separately below. Included studies are summarised in Table 1 and identified factors illustrated using a framework showing the complex relationship between variables. 

## 3. Results

A total of 3278 citations were gathered through database and grey literature searching. After removing 349 duplicates, the search resulted in 2929 citations for screening. After a full-text screening of 103 studies, 26 articles were included in this review (22 quantitative and four qualitative studies) with a total of 13,429 participants (Figure 1).

### 3.1. Characteristics of Included Studies 

The review included 26 studies, which included fourteen randomised controlled trials [16,17,32,33,34,35,36,37,38,39,40,41,47,48]; eight cross-sectional studies [8,15,18,30,41,48,49,50], and four qualitative studies [43,44,45,46]. 

All eligible articles were published in the English language. The majority of the included studies were conducted in the USA [15,16,17,31,32,33,34,35,40,41,43,48,49,50,51] and UK [36,37,44,45,46], and one study was conducted in a developing country, Syria [46]. The remaining studies were conducted in Canada, the Netherlands, and China [8,18,38,39,42]. 

Twenty-five studies were conducted among adult smokers and ex-smoker participants [8,15,16,17,18,31,32,33,34,35,36,37,38,40,41,42,43,44,45,47,48,49,50,51], and one study enrolled adolescent participants who smoked [39]. Of the studies with adult participants, seventeen focused on adherence to NRT in general adult population [8,15,16,17,18,31,32,33,35,36,38,41,42,44,47,50,51], two on HIV-positive adults [40,43], four on pregnant women [34,37,45,46], one on adults with alcohol dependence [49], and one on homeless individuals [48]. The number of participants among the quantitative studies ranged from a low of 101 [31] to a high of 3203 current and ex-smokers [50].

The mean age of participants included in the review ranged from 16.6 years old [39] to 49.9 years old [51]. Among studies enrolling male and female participants, seven studies enrolled more female participants [31,35,38,39,41,42,51], and in seven articles, male participants represented a higher proportion than females [8,33,40,47,48,49,50]. Most of the studies defined adherence to medication as using NRT in accordance with health care provider instructions. Details of the definitions used to assess adherence to NRT for each study are described in Table 1.

### 3.2. COM-B Analysis

The COM-B model (capability, opportunity, and motivation), and its sub-components were used to group factors associated with adherence to NRT among participants who smoke. This systematic review identified a total of 31 factors (Figure 2) associated with adherence to NRT from 26 studies, which were mapped onto the six sub-components of the COM-B model (Table 2).

### 3.3. Psychological Capability

This theme explored the psychological capabilities of participants of the studies to adhere to NRT. Adherence to NRT was found to be associated with the level of education [8]. Completing grade 12 or above was found to be significantly associated with a greater level of adherence to NRT in a study conducted among Chinese current smokers (OR = 2.29, 95% CI of 1.14 to 4.62) [8]. In addition, past experience of using any type of NRT was associated with an improved level of adherence with NRT use instructions provided by the health care provider (Figure 2) [8,45].

Forgetfulness was a significant factor affecting participants’ adherence to taking NRT medication. A study conducted among adolescents in the Netherlands secondary school students reported forgetting to take NRT (38%) as one of the main reasons for non-adherence to NRT. [39] Similarly, 30% of the participants in a study conducted in the U.S documented forgetfulness as a reason for non-adherence to physician instructions. [51] Similarly, a qualitative study conducted in the U.S. among HIV positive individuals reported trouble remembering to use the nicotine patches as one of the factors affecting adherence [43].


*One of the interviewees said, “I’m currently couch surfing…I try to keep the box of patches [nicotine patches] in my suitcase but sometimes I forget.”*
*[43]*

Participants of the study also suggested putting the nicotine patch with other medications or their toothbrush and keeping it in a more visible place as a strategy to avoid forgetfulness [43].


*Another participant said, “I put it where, you know, like the deodorant and the perfumes and the colognes are, which is what I do in some of the medications that I take and they’re right here and I’m going to make sure that I put it on. But that’s how I remember things.”*
*[43]*

Moreover, participants who had a high level of confidence about quitting were found to adhere to NRT better than participants who did not have confidence in themselves [41,48]. However, results of a qualitative study conducted among pregnant women suggested overconfidence in one’s ability to quit without assistance, usually accompanied by a negative belief towards the medication, which can lead to poor adherence to NRT [46].

### 3.4. Physical Capability

This category assessed participants’ physical capability to adhere to NRT. The association between nicotine dependence and adherence is inconsistent, some studies show an inverse relation [31,37,47], while others show a direct relation [16,17,32,42]. Most of the studies used the Fagerstrom test of nicotine dependence (FTND) except one study that used the number of cigarettes per day as a measure of nicotine dependence [37]. The FTND is a standard instrument for assessing the intensity of nicotine addiction. The scale contains six items such as the number of cigarettes per day, difficulty in refraining from smoking in forbidden places, and others. The higher the score the more intense is the nicotine dependence. Scores above 6 indicate a high level of nicotine dependence [10].

Four of the studies included in this systematic review found a higher level of adherence among participants with strong nicotine dependence [16,17,32,42]. Exhaled carbon monoxide is also an indicator of the number of cigarettes one smokes. Higher baseline exhaled carbon monoxide among African Americans was shown to be associated with improved adherence to NRT (OR = 1.22, 95% CI = 1.01 to 1.48) [16].

On the contrary, three studies found an inverse relationship between stronger nicotine dependence and participants’ adherence to NRT [31,37,47]. It was found that people with higher scores of FTND have a lower level of adherence to nicotine patches [31]. For instance, the outcome of a study conducted in Syria, using data from a double-blind randomised trial among adult smokers, concluded that higher FTND scores and a higher number of cigarettes smoked per day were predictors of low adherence to nicotine patches [47]. Higher baseline saliva cotinine level was inversely associated with adherence to NRT [37]. Cotinine is the principal metabolite product produced by the liver when a person smokes tobacco. It easily diffuses from the bloodstream to saliva and is used to estimate the blood cotinine level [52].

Relapse was usually associated with a higher level of nicotine dependence and shown to have an inverse relationship with adherence to NRT [31]. Relapse was found to be one of the most common reasons (10–56% of participants) for premature discontinuation of NRT [15,38,39,51].

The number of years of smoking may be a factor to consider, as it might affect nicotine dependence. Similar to the level of dependence, there are controversies regarding the duration of smoking and its association with adherence to NRT [38,41]. For example, American homeless smokers were more likely to adhere to NRT if they had started smoking at a younger age [48]. However, in another setting, having fewer years of smoking was associated with a higher level of adherence to NRT [49].

Participants who experienced greater levels of withdrawal symptoms had lower adherence to NRT [18,47]. Strategies used to reduce the extent of withdrawal symptoms were associated with improved adherence to NRT. For instance, pregnant women who started NRT within 48 h of quitting were found to be five-fold more adherent to their medication than those who started after 48 h (OR = 5.4, 95% CI = 2.2–12.9) [34]. A secondary analysis of data from a randomised controlled trial showed that prescribing higher doses of nicotine patches instead of lower dose nicotine patches improves adherence rates [36].

NRT may have side effects such as skin irritation, hiccups, loss of appetite, nausea, unpleasant mouth taste, insomnia, irritability, sore throat, and increased blood pressure. Premature discontinuation of NRT due to experiencing side effects was another physical capability factor found to be associated with NRT adherence [15,18,38,39,42,44,51]. The rate of non-adherence due to side effects to the NRT regimen ranged from 15 to 17% in a study conducted in the U.S. [15,51]; from 11 to 19.3% in a study conducted in the Netherlands [18,39], and 14% in a study in Canada [38]. In the reviewed publications, skin irritations were among the factors that led to non-adherence to NRT use (Figure 2) [44,45].


*One participant said, “Regardless of where I put it…my skin was still itching through the patch, so I’d scratch right on the patch.”*
*[44]*

In addition, the taste of oral NRT can be a factor for non-adherence, which is illustrated in a qualitative study.


*One consumer using oral NRT said, “Yeah, totally delayed cos I keep saying ‘oh I don’t want to taste it yet, I’ll give it another ten minutes you know, or I’ll give it a bit longer’. It is delaying it cos you think I’m not looking forward to the taste of it so I’ll just wait a bit longer.”*
*[45]*

In addition, among pregnant women, increased intensity of morning sickness was a deterrent to adherence to NRT. [45]


*A pregnant woman said, “I was just more worried about the side effects obviously because I’m quite early on in pregnancy, and especially with morning sickness anyway, I didn’t know that it [NRT] would cause—obviously [it] made me feel more nauseous and [I] vomited quite a few times when I had the gum. So just would have been nice to have a heads up about it that it makes you feel sick”.*
*[45]*

### 3.5. Motivation

Studies consistently reflected the direct relationship between the level of motivation to quit smoking and adherence to NRT [31,41,43,44,48]. When attempting to quit cigarette smoking, motivation affects an individual’s decision making and goal setting. Having a low level of motivation to use NRT as well as not being ready to quit smoking reduced the level of adherence [41,48]. Motivated participants are more likely to have the energy to take the NRT as prescribed. For instance, in one study, the rate of adherence was found to be 57.2% among motivated participants and 37.1% among individuals with less level of motivation to quit smoking [41]. Moreover, participants of qualitative study demonstrated how motivation affects adherence (Figure 2) [43].


*One participant said, “… I know it’s helping me out. I haven’t missed a dose because I’m motivated.”*
*[43]*

Motivation has two sub-components: Reflective motivation encompasses individuals’ conscious decision, plan, and evaluation of the problem and the NRT. Whereas, automatic motivation includes impulses, emotions, habits, and desire to see positive health or other outcomes [20]

### 3.6. Reflective Motivation

The reflective motivation category includes factors that affect an individual’s evaluation of NRT and quitting, planning, and conscious decision-making processes. Adherence to NRT was found to be poor among participants who believed NRT has no effect in supporting one’s quit attempt. For example, in a study conducted in the Netherlands, more than one-third of adolescents believed NRT had no effect and discontinued the treatment prematurely [39]. Similarly, 14% of adult participants in a study conducted in the U.S. discontinued NRT because they believed it was not helpful [15]. Additionally, throughout the treatment process mistakenly assuming that NRT was no longer necessary was a driver of poor adherence. A large multi-national survey from four countries (Australia, Canada, USA, and UK) found that 16.3% of participants discontinued the medication prematurely because they believed it was no longer needed [42].

Concern about the safety of NRT was one of the main factors affecting participants’ reflective motivation to use the medication. Participants who believed NRT was not safe tended to discontinue the medication prematurely or utilised lower doses of NRT than the recommended dose [50]. Thus, hesitancy to use NRT due to belief about safety and efficacy was associated with poor adherence [38]. Some participants also believed that taking NRT may increase their addiction to cigarettes [45].


*One person stated that, “Well, all they keep saying is you know it gets rid of the toxins, you still get the nicotine but it gets rid of the toxins, this, that and the other and it’s just in that the nicotine you take it in. The nicotine itself is what makes it addictive, so to me the more nicotine that you’re taking in any way, the more you’re going to want to smoke or you know you’re going to need that nicotine.”*
*[45]*

The perception of smokers and ex-smokers that following health care providers’ instructions was an easy task was associated with better adherence to NRT [41].

When individuals failed in their attempt to quit smoking several times, it was associated with a decreased motivation for further attempts. It has been shown that the higher the number of previous quit attempts, the lower adherence to NRT and vice versa (Figure 2) [16,34].

### 3.7. Automatic Motivation

This category includes factors that affect adherence to NRT by influencing an individual’s emotions, impulses, and unconscious associations with the performance of the desired behaviour, e.g., daily routines.

In this automatic category, one of the main factors affecting adherence to NRT was found to be the mental health status of the individual. Participants who were not diagnosed with depression had more than double the level of adherence to NRT compared with participants who were diagnosed with depression (OR = 2.48, 95% CI, 1.14 to 5.39) [17]. Similarly, a study among homeless smokers showed a lower level of adherence among participants with depressive symptoms [48].

Individuals who were experiencing a higher level of stress had a lower level of adherence to NRT [38]. However, awareness of stress during a quit attempt increased the odds of being adherent to NRT [16]. Similarly, individuals with an anxiety disorder were found to be less adherent than those without anxiety symptoms [35].

Alcohol consumption was associated with an increased level of smoking. It was shown that individuals with alcohol dependence problems found adhering to NRT challenging [49].

In addition to this, fearing dependence to NRT [18] and being able to incorporate the habit of using the medication with daily routines impinge on adherence to NRT [43]. Linking NRT use with other daily routine activities seemed to improve the level of adherence. Being part of the daily routine motivates participants in adhering to NRT (Figure 2) [43].


*A participant said, “It’s a routine. I put it on after I shower in the morning and put the patches next to my medication.”*
*[43]*

Similarly, HIV-positive Latino participants in a qualitative study illustrated how they associated taking the NRT with their other medications [44].


*Another interviewee stated, “I leave it [the patches] next to my medication and remember to take it at the same time each day.”*
*[44]*

### 3.8. Physical Opportunity

This subcomponent describes how the physical environment affects an individual’s level of adherence to NRT (Figure 2). The cost of NRT was the main environmental factor that was associated with the level of adherence to NRT, since it affected an individual’s motivation to take NRT as prescribed by the health care provider. For instance, data from the Colorado state tobacco survey demonstrated that 5% of participants discontinued the NRT because they could not afford the cost. Moreover, participants who discontinued the treatment due to cost issues had less intention to use NRT in a future quit attempt. [15] Likewise, 7% of participants in a study conducted in the U.S. discontinued the nicotine patch due to cost-related issues [51].

Although NRT is an over-the-counter product approved by the Food and Drug Administration (FDA) in the U.S., its price may be prohibitively high for most minors, thus discouraging effective use of NRT [53].

It is suggested that factors such as gender and age have an association with adherence to NRT. Being a male and of older age was associated with better adherence to NRT [8,33,35,49]. For instance, one study showed that males are more than twice as likely to be adherent than females (OR = 2.38, 95% CI, 1.25 to 4.55) [8]. The same study illustrated that older age is associated with a greater level of adherence to NRT (OR = 2.45, 95% CI, 1.29 to 4.64) [8].

### 3.9. Social Opportunity

This component of the model consists of factors outside of the individual that makes the performance of the behaviour possible. Having a better social network and support affects adherence to NRT both directly and indirectly through influencing one’s motivation to quit smoking (Figure 2). A randomised trial conducted among HIV-positive smokers in the U.S. evaluated the effect of social support on adherence to NRT. The study utilised a modified version of the Important People and Activities Instrument. The findings indicated that having greater social support network contact is significantly associated with a higher level of adherence to nicotine patch [40]. On the contrary, one study reported a lower level of adherence to NRT among participants who lived with a child or children. Living with a child or children may lead to forgetting to take the NRT and increased psychological stress due to the responsibility it brings to the individual [35].

## 4. Discussion

This review identifies factors using a predefined behavioural model, the COM-B model, which is at the hub of the BCW that was effectively used to design and implement behavioural change strategies. Hence, the intervention functions and policy categories in the BCW are used to recommend and discussed the findings [54].

Strategies that aimed at addressing intervention functions of the BCW such as enablement, education, and training were found to be effective in improving psychological capability of an individual towards adherence to smoking cessation medications [19,55]. For instance, a double-blind randomised controlled trial conducted in the UK evaluated the effect of a mobile application (NRT2Quit) among adult smokers. The application provided comprehensive information about quitting and NRT, daily tips as well as a reminder about the medications. The intervention group had a higher level of treatment adherence and 4 weeks biochemically verified smoking cessation rate (25%) compared to the control group (8%). This study supports our findings on the importance of an individual’s ability to remember and comprehend the necessary information and instructions about NRT use [56].

In this review, except for the level of nicotine dependence, all other factors mapped under physical capability were consistently associated with either high or low level of adherence to NRT. Studies that reported an inverse relationship between the level of dependence and adherence to NRT also reported relapse to smoking and dropout from the study, which may have contributed to the inverse relationship between the two variables [31]. Prescribing higher doses of NRT and early initiation were found to improve the level of adherence to NRT [36]. This can be explained by the fact that higher doses are more effective in alleviating withdrawal symptoms from quitting than low dose preparations leading to better adherence. This is represented as enablement in the intervention function and guideline in the policy category of the BCW [54].

Furthermore, withdrawal symptoms were one of the most commonly reported causes for nonadherence in the category of physical capability. When attempting to quit smoking, experiencing greater levels of withdrawal symptoms could become a barrier to abstinence, which also applies to NRT aided smoking cessation [57]. This explains why, when participants attempt to quit smoking “cold turkey” (unaided) or delay initiation of NRT, they often fail to quit or adhere to NRT [38]. To minimise withdrawal symptoms, the dose of NRT should be adjusted according to one’s dependence and severity of withdrawal symptoms [58,59].

Medication-related beliefs and expectations were the main factors categorised under the reflective motivation subcomponent of the COM-B model. A randomised factorial study also found a significant positive effect of additional medication-related face-to-face counselling and automated phone calls directed to improve knowledge about NRT [60]. Establishing realistic expectations are also vital during quit attempt. A study conducted by Tucker et al. in 2017 [61] reported a higher adherence rate to NRT among participants who received additional information on the extent of withdrawal symptoms and urges and how NRT would reduce them in order to develop realistic expectations.

Motivation improves once ability to cope up with withdrawal symptoms and improves the appropriate consumption of NRT [62]. Psychological symptoms and alcohol use reduced the rate of motivation to quit and adherence to NRT. This could be explained by the risk of resuming smoking among participants who have developed the above mental issues [62,63]. There is a close link between smoking and drinking. Alcohol intake may trigger cravings to smoke cigarettes [64]. Alcohol may affect adherence to NRT by increasing the risk of resuming smoking [65]. As alcohol intake is often cited as a major precipitant of smoking relapse [66], current clinical guidelines for smoking cessation also suggest reducing or avoiding drinking during quit attempt [67]. The rewarding effect of smoking is enhanced in smokers with alcoholic disorders. Hence, smokers with alcohol use disorders tend to experience intense withdrawal symptoms and craving leading to resuming smoking and medication nonadherence [68].

The rate of adherence to NRT was improved among clinical trial participants for whom associating medication with regular activities such as taking other medications, with meals, and watching preferred TV shows were achieved with the help of therapists [69]. Being able to identify and prevent personal triggers reduces the risk of relapse and adherence to NRT. This finding is similar with a randomised trial that aimed at identifying and avoiding temptations to smoke, which helped improve adherence at eight weeks as well as self-reported abstinence rates [61].

Factors categorised under opportunity component of the COM-B model can be improved by addressing the fiscal measures and legislation elements in the BCW that advocate access to NRT [54]. Although most countries subsidise the cost of NRT and provide the medication over the counter, it may still be difficult to get a preferred or combination of forms of NRT. NRT may only be prescribed for a shorter period, which may not be enough to support cessation leading to premature discontinuation of NRT [70]. Greater social support increases adherence to nicotine patches, which may be explained by the effect of having someone to remind the participant to take the medications, motivate them to stay quit, and provide financial support to refill medications [71]. Additionally, having greater social support may improve the mental health status of an individual during a quit attempt [72].

### 4.1. Strength and Limitations of the Study

Our search strategy was broad and comprehensive and was developed with the help of an experienced librarian. As only one study was conducted in a developing country, caution should be taken during interpreting the findings especially in developing countries; there were also some inconsistencies in the direction of the association between some factors. Moreover, the COM-B model did not distinguish intentional and unintentional nonadherence. Despite these limitations, this systematic review is the first to evaluate enablers and facilitators of adherence to NRT using the theoretical framework of the COM-B model.

### 4.2. Implications for Policy, Research, and Practice

Health care providers are recommended to provide adequate information about withdrawal symptoms and NRT, address safety concerns, and establish realistic expectations. Individuals are recommended to boost their motivation and mental health by having social connections, physical activity, and managing stress. It is also recommended for patients to visit health care providers, smoking cessation support services, or access a Quitline for psychological support and counselling.

Subsidisation of smoking cessation medications and care are advocated and recommended to improve adherence to NRT and the success of smoking cessation attempts. It is recommended to advocate policies and strategies that can enhance motivation to quit smoking through the promotion of the health benefits of smoking cessation and smoking cessation supports.

Additionally, as the review illustrated the importance of factors at multiple levels, implementation and evaluations of trials addressing multiple components of the COM-B level are recommended.

More studies, especially among special population groups, such as pregnant women, individuals with psychiatric disorders, and socially disadvantaged individuals, are recommended to develop more tailored approaches. In addition, as gender is an important factor for adherence and successful quitting, future research should include gender-based analysis.

## 5. Conclusions

The current review demonstrated the importance of personal, social, and environmental factors affecting adherence to NRT using a comprehensive predefined theoretical framework (COM-B model). Most of the identified factors were mapped under the category of the reflective motivation component of the COM-B model followed by physical capability, and automatic motivation. Hence, reflective motivation is the most crucial element for adherence to NRT out of the six sub-components of the COM-B model. However, it should be noted that all sub-components of the COM-B model are essential in moderating an individual’s behaviour concerning adherence to NRT. For instance, motivation cannot provide opportunity, and if these are missing, then according to the COM-B model, behaviours have to be engaged with to increase or seek out opportunities, which may be beyond an individual’s control.

## Figures and Tables

**Figure 1 ijerph-17-08895-f001:**
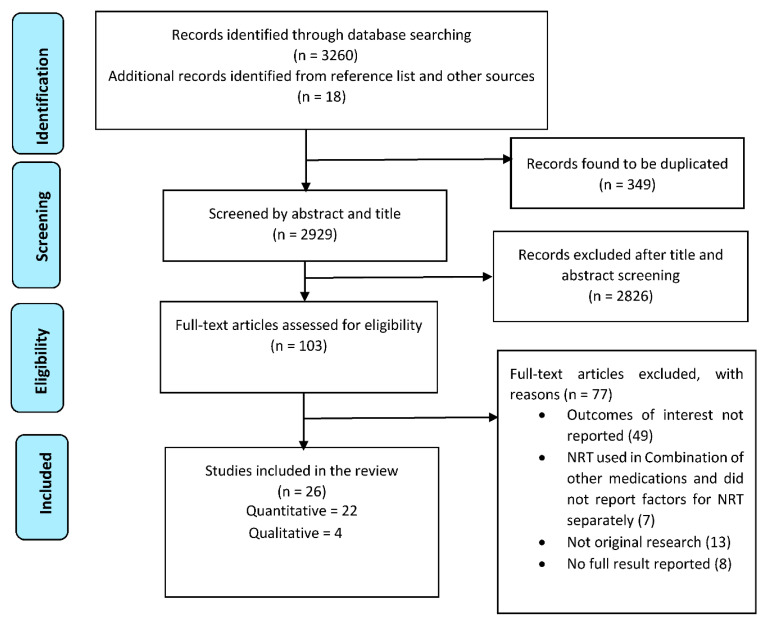
PRISMA flow diagram of studies included in the review.

**Figure 2 ijerph-17-08895-f002:**
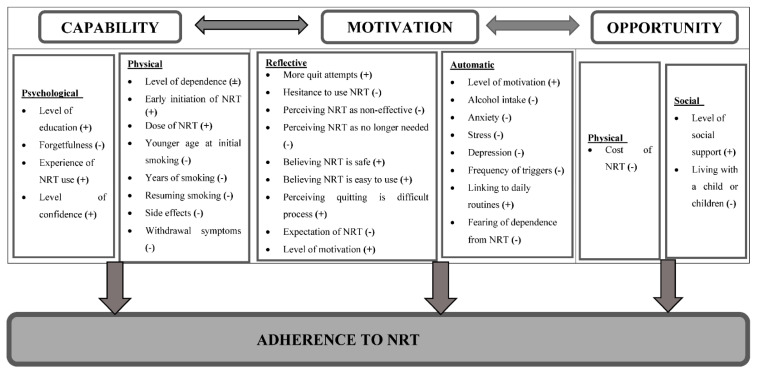
Framework illustrating factors affecting adherence to NRT using the COM-B model. “(+)” indicates direct relationship, “(-)” indicates inverse relationship, and “(±)” indicates inconsistent relationship across studies.

**Table 1 ijerph-17-08895-t001:** Overview of studies included in the systematic review.

Author, Year of Publication, References	Country of Study	Study Design and Sample Size	Participants	Definition of Adherence	Main Outcomes on Factors Associated with Adherence	Quality
Alterman, 1999 [31]	USA	Cross-sectional, 101	Males or non-pregnant females between the ages of 18 and 65, who met DSM-IV criteria for nicotine dependence and reported at least one previous failed quit attempt.	Patch adherence was obtained by counting dispensed and used patches by the research technician.	Greater dependence was associated with less patch use, indicating that subjects smoking more cigarettes were less patch adherent. A positive relationship between greater motivation and more patch use was demonstrated.	62%
Balmford, 2010 [42]	Australia, USA, UK, Canada	Cross-sectional, 981	Adult smokers or recent ex-smokers who reported having made a quit attempt in the previous year and reported using NRT to help them quit.	Completion of a course of treatment was defined as use for 8 weeks, with those who terminated before this cut point considered to have stopped prematurely.	Relapse back to smoking was the most common reason for discontinuation of medication reported by 41.6% of respondents. Side effects (18.3%) and believing that the medication was no longer needed (17.1%) were also commonly reported.	87%
Ben Taleb, 2015 [47]	Syria	RCT, 269	Adult smokers in the age group of 18 to 65 years old.	Participants were asked whether they had followed treatment instructions to use one patch every day over the past week. Adherence to patch use as responding “yes” to this question during at least 5 of the 6 weeks (>80%).	Participants who smoked a greater number of cigarettes per day at baseline (OR = 0.97; 95% CI = 0.94–0.99) and had higher withdrawal symptoms (OR = 0.97; 95% CI= 0.95–0.98) were less likely to adhere to patch usage.	92%
Berg, 2013 [32]	USA	RCT, 202	Adult smokers of 18 years of age or above.	Calculated adherence level as the number of patches used (80% adherence as adherent; <80% adherence was considered nonadherent).	Predictors of patch adherence included greater prior smoking levels and more quit attempts in the past.	69%
Bowker, 2016 [45]	UK	Qualitative study, 14	Pregnant smokers are prescribed with NRT.	Pregnant women not using NRT as it was recommended by the health care provider.	Four main themes were identified: expectations of NRT, the experience of using NRT, safety concerns, and experience of using e-cigarettes. Low NRT adherence is associated with believing the risk of smoking and NRT as comparable.	80%
Burns, 2008 [15]	USA	Cross-sectional, 366	Currently smoked every day or some days or had quit within 365 days of the interview date, tried to quit within 365 days of the interview, and used NRT in the most recent quit attempt.	NRT usage as prescribed by a health care provider instruction.	Discontinuing NRT factors were resuming smoking (34%), side effects (17%), NRT not helping with quitting (14%), quitting smoking (10%), and cost (5%).	75%
Cooper, 2004 [33]	USA	RCT, 619	Adults who smoked at least 10 cigarettes per day, had been smoking for at least a year.	Based on patch use during the 6 week treatment period, participants were categorized as fully adherent (used ‘‘all of the patches’’), partially adherent (used ‘‘most or some of the patches’’), or nonadherent (used ‘‘a bit or none of the patches’’).	Not dropping out of the study and intensive treatment (compared to the standard care).	69%
de Dios, 2016 [40]	USA	RCT, 444	Participants were eligible if they were: (1) seropositive for HIV, (2) 18+ years of age, (3) currently smoking (≥5 cigarettes/d for the past 3 months).	Adherence was measured using retrospective self-reports of NRT patch collected at each follow-up visit.	Greater social support network contact was associated with higher levels of nicotine patch adherence.	85%
Fish, 2009 [34]	USA	RCT, 104	Pregnant women, GA 13–25 weeks, smoked at least 100 cigarettes in their lifetime, currently smoking five cigarettes per day.	Total days of nicotine patch use per week for the follow-up period.	Using NRT as directed in the first 48 h (OR = 5.4, 95% CI = 2.2–12.9, *p* = 0.0002) and having made a previous quit attempt (OR = 2.9, 95% CI = 1.1–7.6, *p* = 0.04) were the strongest predictors of longer NRT use.	85%
Handschin, 2018 [35]	USA	RCT, 440	18 years of age or older, report smoking at least 10 cigarettes per day and had to express an interest in quitting smoking.	Adherent (≥80% of daily patch use) and non-adherent participants (<80% of daily patch use.	In a logistic regression model, being female, living with a child or children, and higher self-reported anxiety symptoms were predictive of lower patch adherence.	85%
Hollands, 2013 [36]	UK	RCT, 633	All participants were prescribed a nicotine patch and oral NRT dose.	The proportion of all NRT prescribed consumed each day, averaged over the 4 week treatment period.	Prescribing higher doses of patch and oral NRT was associated with higher mean daily consumption of NRT.	77%
Hood, 2013 [17]	USA	RCT, 147	Participants were adult women living in Ohio on NRT for smoking cessation.	Patch adherence was dichotomised into 7 weeks or less versus >7 weeks to distinguish between participants who received close to the recommended 8 weeks of patches from those who did not.	Depressive symptoms and low nicotine dependence were associated with lower patch adherence, while the high poverty-to-income ratio was associated with high responsiveness.	85%
Kim, 2019 [41]	USA	RCT, 623	Participants needed to be willing to quit smoking in the next 30 days, at least 18 years old, smoking at least 5 cigarettes per day for the previous 6 months.	Daily patch use was coded as binary (0 = used patches 6 or fewer days in the past week and 1 = used patches every day for the past week) and mean daily mini-lozenge use was coded as an ordered categorical variable.	Greater baseline dependence predicting greater medication use. Greater quitting motivation and confidence and believing that smoking cessation medication was safe and easy to use were associated with greater adherence.	77%
Kushnir, 2017 [38]	Canada	RCT, 421	Adult current daily smokers who had smoked at least 10 cigarettes per day.	The number of nicotine patches used was assessed at an 8 week follow-up (end-of-treatment) survey by asking respondents “how much of the nicotine patches did you use?”, with the response options of “none”, “some”, “all”.	The most common reasons for using only some of the 5 weeks of nicotine patches were delayed initiation, side effects, and discontinuation of use due to stress. Among individuals who have not used any of the nicotine patches, the most common reasons were not being ready to quit, stress, and hesitance to use because of the misperception of nicotine patch effects or side-effects.	85%
Lam, 2004 [8]	China	Cross-sectional, 1051	Adult current smokers using NRT for smoking cessation.	Self-reported use of NRT daily for at least 4 weeks during the first 3 months.	Higher education, the experience of NRT use, perceiving quitting as more difficult, and willingness to pay were significant predictors of adherence.	87%
McDaid, 2020 [46]	UK	Qualitative study, 18	Pregnant or recently pregnant women in England and Wales who gave birth within 6 months.	NRT usage as prescribed by a health care provider’s instruction during pregnancy.	NRT adherence was found to be associated with pregnant women’s preference for quitting unassisted, unrealistic expectations, overconfidence, safety concerns, side effects, and capability to use.	80%
Ojo-Fati, 2016 [48]	USA	RCT, 430	Being currently homeless, smoked at least 5 cigarettes per day, smoked at least 100 cigarettes in a lifetime, and smoked at least one cigarette every day.	Adherence was defined as a total score of zero in a modified Morisky adherence scale at the end of NRT treatment (8 weeks).	After adjusting for confounders, smokers who were depressed at baseline (OR = 0.58, 95% CI, 0.38–0.87, *p* = 0.01), had lower confidence to quit (OR = 1.10, 95% CI, 1.01–1.19, *p* = 0.04), were less motivated to adhere (OR = 1.04, 95% CI, 1.00–1.07, *p* = 0.04), and were less likely to be adherent to NRT. Further, the age of initial smoking was positively associated with adherence status (OR = 0.83, 95% CI, 0.69–0.99, *p* = 0.04).	92%
Okuyemi,2010 [16]	USA	RCT, 755	African American light smokers (defined as smoking ≤10 cigarettes/day).	Adherence to gum was defined as using greater than or equal 75% of the total prescribed gum usage during the 8 weeks of treatment with gum.	Having more quit attempts in the past year (OR= 1.04, 95% CI = 1.01 to 1.07), higher baseline exhaled carbon monoxide (OR = 1.22, 95% CI = 1.01 to 1.48), and higher perceived stress (OR = 1.12, 95% CI = 1.03 to 1.22) increased the likelihood of adherence to nicotine gum.	92%
Rojewski, 2016 [49]	USA	Cross-sectional, 843	18 years of age and meet hazardous drinking criteria as defined by the National Institute on Alcohol Abuse and Alcoholism (NIAAA).	NRT use was assessed by self-report at the 7 month follow-up. Participants were asked to select which category best described their level of medication use: (1) all of it, (2) about half of it, (3) less than half of it, or (4) none of it. NRT use was coded as follows: used all NRT = 1, used some NRT = 2 (collapsed half and less than half together), used none = 3.	Those who used all of the NRT had been smoking for a fewer number of years (22.8 ± 12.8), reported a lower percentage of heavy drinking days at baseline (11.6%), and were more likely to complete the second counselling session (38.4%).	87%
Scherphof, 2014 [39]	The Netherlands	RCT, 265	Participants were allowed to participate if they were 12 years up to and including 18 years old, they smoked at least seven cigarettes a day, they were motivated to quit smoking.	The number of days participants had used the patches.	Reasons for non-adherence were having the feeling that the patches had no effect (38.0%), forgetfulness (37.4%), experiencing side effects (19.3%), and quitting smoking (10.2%).	77%
Shadel, 2016 [43]	USA	Qualitative study, 35	At least 18 years of age, Latino, HIV-positive, smoked at least 5 cigarettes per day for at least the last 20 days, and had used the nicotine patch during any past quit attempt.	Consumption of NRT as prescribed by the health care provider.	Consistent use of the nicotine patch was associated with maintaining high motivation for use (i.e., not necessarily motivation to quit, but motivation to continue patch use); linking its use with established daily routines (e.g., with taking other medications, with brushing teeth); and maintaining realistic expectations for patch efficacy (e.g., that users may still experience some level of craving and/or withdrawal).	80%
Shiffman, 2008 [50]	USA	Cross-sectional, 3203	Adult smokers or ex-smokers who had quit within the last year.	Participants asked about the length of time they used the product and the average number of pieces/patches that they used per day when they were using the product/s.	Adherence was associated with believing stop-smoking products with nicotine are just as harmful as cigarettes, having concerns about the safety of NRT, not believing NRT to be efficacious.	87%
Vaz, 2016 [37]	UK	RCT, 1050	Pregnant women between 12 to 24 weeks of gestation from 1050 pregnant trial SNAP participants.	At 1 month, participants could report using patches for a maximum of 28 days and at delivery for a maximum of 56 days; adherence was measured with respect to these values.	Adherence during the first month was associated with lower baseline cotinine concentrations.	92%
Wiggers, 2006 [18]	The Netherlands	Cross-sectional, 174	Adults smoked > 5 cigarettes a day, received free patches, and intensive instructions from nurses.	Using the prescribed patches for 7–8 weeks as prescribed.	Low adherence was associated with not wanting to use the patches at the same time (13%), being allergic to NRT (11%), having doubts about the effectiveness of NRT (9%), fearing becoming dependent to the patches (7%), or the patches falling off (6%).	75%
Wright, 2018 [44]	UK	Qualitative study, 40	Individuals had to smoke at least 10 cigarettes a day.	Participants were requested to take their NRT as prescribed for 4 weeks after their quit date.	Adherence to NRT is associated with the presence of side effects, forgetfulness, or practical difficulties.	70%
Yingst, 2015 [51]	USA	Follow-up cross-sectional study, 201	Current daily smokers recruited from the Penn State Hershey Medical Centre and surrounding family medicine outpatient centres.	Adherence to the directed use of the nicotine patch was measured by the number of self-reported days, of 28 days, the patch was worn during the quit attempt in treatment. Participants were considered adherent if the patch was worn all 28 days and non-adherent if the nicotine patch was worn less than 28 days.	Reasons for non-adherence were forgetting to put the patch on (30%), not liking the experienced side effects (15%), resuming smoking (10%), and difficulty affording the cost of the patches (7%).	87%

DSM—The Diagnostic and Statistical Manual of Mental Disorders, NRT—Nicotine replacement therapy, RCT—Randomised controlled trial, SNAP—Smoking, Nicotine And Pregnancy.

**Table 2 ijerph-17-08895-t002:** Table illustrating covered components of the capability, opportunity, motivation, and behaviour (COM-B) across studies.

Study	Psychological Capability	Physical Capability	Reflective Motivation	Automatic Motivation	Physical Opportunity	Social Opportunity
Alterman [31]		✓		✓		
Balmford [42]		✓	✓			
Ben Taleb [47]		✓				
Berg [32]		✓	✓			
Burns [15]		✓	✓		✓	
Cooper [33]		✓				
de Dios [40]						✓
Fish [34]		✓	✓			
Handschin [35]				✓		✓
Hollands [36]		✓				
Hood [17]		✓		✓		
Kim [41]	✓	✓	✓	✓		
Kushnir [38]		✓	✓	✓		
Lam [8]	✓		✓			
McDaid [46]	✓	✓	✓	✓		
Ojo-Fati [48]	✓	✓	✓	✓		
Okuyemi [16]		✓	✓	✓		
Rojewski [49]		✓				
Scherphof [39]	✓	✓	✓			
Shiffman [50]			✓			
Vaz [37]		✓				
Wiggers [18]		✓	✓	✓		
Yingst et al. [51]	✓	✓	✓		✓	
Bowker [45]	✓		✓	✓		
Shadel [43]			✓	✓		
Wright [44]	✓	✓		✓

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
