# Peer review of "Barriers and Facilitators of Adherence to Nicotine Replacement Therapy: A Systematic Review and Analysis Using the Capability, Opportunity, Motivation, and Behaviour (COM-B) Model"

_ijerph, 2020, doi:10.3390/ijerph17238895_

Round 1

Reviewer 1 Report

This is a systematic review aiming to understand the barriers and facilitators of adherence to nicotine replacement therapy using the COM-B (Capability, Opportunity, 4 Motivation, and Behaviour) model. Overall, the study provides a great overview on the factors affecting NRT adherence. The manuscript is well-written with the following suggestions that need to be improved or addressed:

  • In the background (line 43-50), some of the literature were studying adherence to Varenicline, which is quite different than NRT, and thus should be removed or noted in the review since the mechanisms of action, delivery methods between the two are quite different and so does the adherence.
  • Review articles include studies using multiple smoking cessation medications if they reported factors associated with NRT separately. What are your screening process to address the issue when the adherence to NRT might be confounded by the other medication? Are they using both or more than one method?
  • The paper did not explain what the reasons to exclude 2,826 papers from 2,929 were (Figure 1). Were there any issues in the first step of database searching so that most of the search results were not relevant? The authors might need to address this issue by using a different strategy.
  • Why clustered studies from Canada, Netherlands, and China, while single out a study from Syria (line 164-165)? Usage behavior might be quite different between those countries if the authors want to provide a better picture for the audience on where the data was from.
  • Typo on page 13 “capablity”.
  • Reference 51 did not mention cotinine as the principal metabolite produced by the liver. Please replace with a correct reference.
  • Line 373-382 seems to be part of the result and not discussion.
  • Revise line 409-410: “There is a close link between smoking and drinking were alcohol is a trigger for smoking.”

Author Response

Reviewer 1

Thank you for your thorough review of our manuscript.

  • Question - In the background (line 43-50), some of the literature were studying adherence to Varenicline, which is quite different than NRT, and thus should be removed or noted in the review since the mechanisms of action, delivery methods between the two are quite different and so does the adherence.
  • Reply - Details of the referenced review are added as suggested “The review included studies that assessed any type of smoking cessation medications such as varenicline, bupropion and NRT.” (Page number - 2, Line number - 43)
  • Question - Review articles include studies using multiple smoking cessation medications if they reported factors associated with NRT separately. What are your screening process to address the issue when the adherence to NRT might be confounded by the other medication? Are they using both or more than one method?
  • Replay - Among studies that evaluated adherence in all types of smoking cessation medications, we include if the study evaluates adherence to only NRT users separately. For example, if a study has 500 participants and 200 of them were using only NRT and reported factors for the 200 participants separately, we included the findings for the 200 participants. However, if analysis were done all together, we excluded the study. If participants used NRT and other smoking cessation drugs at the same time the study is excluded. (Page number - 3, Line number - 90) In the manuscript, it is written as “if the study identified factors for only NRT users separately”
  • Question - The paper did not explain what the reasons to exclude 2,826 papers from 2,929 were (Figure 1). Were there any issues in the first step of database searching so that most of the search results were not relevant? The authors might need to address this issue by using a different strategy.
  • Replay - It is corrected as citations excluded after title and abstract screening. (Page number - 9, figure - 1)
  • Question - Why clustered studies from Canada, Netherlands, and China, while single out a study from Syria (line 164-165)? Usage behavior might be quite different between those countries if the authors want to provide a better picture for the audience on where the data was from.
  • Replay - In addition to provide information on where the data come from, we aim to illustrate the scarcity of studies from developing countries by indicating only one study from Syria and further stated as a limitation during utilisation of the findings for developing country as there is only one study included.
  • Question - Typo on page 13 “capability”.
  • Replay - Typo corrected (Page number - 13, figure - 2)
  • Question - Reference 51 did not mention cotinine as the principal metabolite produced by the liver. Please replace with a correct reference.
  • Replay - Reference replaced with “Benowitz NL, Hukkanen J, Jacob P 3rd. Nicotine chemistry, metabolism, kinetics and biomarkers. Handb Exp Pharmacol. 2009;(192):29-60. doi:10.1007/978-3-540-69248-5_2” (Line number – 621, reference number 51)
  • Question - Line 373-382 seems to be part of the result and not discussion.
  • Replay - The main aim of the paragraph is to demonstrate why we rely on the BCW to discuss and recommend our findings and how different intervention functions of the BCW improved psychological capability barriers of adherence and quitting. The referenced studies are used to justify why the factors mapped under the category of psychological capability are important and can be modified using the BCW intervention functions.
  • Question - Revise line 409-410: “There is a close link between smoking and drinking were alcohol is a trigger for smoking.”
  • Replay - Revised as “There is a close link between smoking and drinking. Alcohol intake may trigger craving to smoke cigarette.” (Page number - 17, Line number - 414)

Reviewer 2 Report

The manuscript by Mersha et. al. summarizes the outcome of the multiple recent field research work related to nicotine replacement therapy (NRT). The strength of this this study lies in the use of PRISMA guidelines and literature searches. The author rely on the literature searching and citation screening using Five electronic databases and the keywords.

The manuscript does a reasonably good job of introducing the topic of the nicotine replacement therapy. I find the methodology used in this study is clearly stated and well described in the paper. The  quality of the writing is overall good, some minor suggestions are made in the following section to improve this paper. In my knowledge, I do not find this work is recent and noble in the scientific community, making a review paper appropriate.

Minor comments:

Line 122: I think it’s a typo “softwear”, should be “software”

Figure 1: I suggest, make lines appealing and arrow straights, connected and organize without overlapping the boxes.

In result section:

Line 178: “… table 1. [table 1]” I did not get why you have two table 1. Similar issues with figures.

Line 199: The quotation looks good, however it does not have any connection or connection to previous sentence.  You should have some connection word and add the quotes. For example  you can write it as : ‘One of the volunteer say “ I’m currently couch suffering ….. but I sometime forgot” [43]’.  

Line 202: full stop should be after your reference number.

Line 204: its similar to what I mentioned earlier for line 199

Line 210: full stop should be after your reference number.

Line 264:  “… [figure 2]”, this is inappropriate it should be “ (figure 2) [43,44].”

Line 265, 269, 274, 285,335:  its similar to what I mentioned earlier for line 199

 I found few places, you have similar problem I mentioned for the line 210. So, please check the full stop which should be end of the sentences after the references, figure not before reference or figures referred.

Author Response

Reviewer 2  

Thank you for a detailed review of our manuscript

  • Question - Line 122: I think it’s a typo “softwear”, should be “software”
  • Replay - Typo corrected to software (Page number - 3, Line number - 124)
  • Question - Figure 1: I suggest, make lines appealing and arrow straights, connected and organize without overlapping the boxes.
  • Replay - Figure modified and adjusted (Page number - 9, figure - 1)
  • Question - Line 178: “… table 1. [table 1]” I did not get why you have two table 1. Similar issues with figures.
  • Replay - Repetition deleted (Page number - 10, Line number - 180)
  • Question - Line 199: The quotation looks good, however it does not have any connection or connection to previous sentence.  You should have some connection word and add the quotes. For example you can write it as : ‘One of the volunteer say “ I’m currently couch suffering ….. but I sometime forgot” [43]’.  
  • Replay - Corrected using connecting words

One of the volunteers said, “I’m currently couch surfing…I try to keep the box of patches [nicotine patches] in my suitcase but sometimes I forget.” [42] (Page number - 12, Line number - 201)

  • Question - Line 202: full stop should be after your reference number.
  • Replay - Corrected throughout the manuscript
  • Question - Line 204: its similar to what I mentioned earlier for line 199
  • Replay - Corrected as

One participant said, “I put it where, you know, like the deodorant and the perfumes and the colognes are, which is what I do in some of the medications that I take and they're right here and I'm going to make sure that I put it on. But that's how I remember things.” [42] (Page number - 12, Line number - 205)

  • Question - Line 264:  “… [figure 2]”, this is inappropriate it should be “ (figure 2) [43,44].” 
  • Replay - Corrected as “(figure 2) [43, 44].” (Page number - 14, Line number - 266)
  • Question - Line 265, 269, 274, 285,335:  its similar to what I mentioned earlier for line 199
  • Replay - Connecting words added (Page numbers – 14 and 15, Line numbers – 268,272,277,290,310)
  • Question - I found few places, you have similar problem I mentioned for the line 210. So, please check the full stop which should be end of the sentences after the references, figure not before reference or figures referred.
  • Replay - Full stop moved to after references throughout

Reviewer 3 Report

This is a relevant systematic review and critical analysis on the factors that may foster or undermine adherence to NRT based in a theoretical framework for behaviour change, the COM-B model.

The methodology for the systematic review is sound and the critical analysis is underpinned in comprehensive  knowledge  and understanding both of behaviour change and smoking cessation and NRT use.

My recommendation is to include a brief reference for the need to develop more studies among special populations of smokers, such as adolescents, pregnant women,  psychiatric patients, socially disadvantaged groups, etc. It is important to target these groups to understand their specificities and develop more tailored approaches. Gender is an important determinant of both adherence and NRT success. Further research should include gender analysis.  

Author Response

Reviewer 3

Thank you for reviewing our manuscript

  • Question - My recommendation is to include a brief reference for the need to develop more studies among special populations of smokers, such as adolescents, pregnant women,  psychiatric patients, socially disadvantaged groups, etc. It is important to target these groups to understand their specificities and develop more tailored approaches. Gender is an important determinant of both adherence and NRT success. Further research should include gender analysis.  
  • Replay - We agree that more studies are necessary especially for a special group of population and we include the following recommendation for future research directions. “More studies, especially among special population groups such as pregnant women, individuals with psychiatric disorders, and socially disadvantaged individuals, are recommended to develop more tailored approaches. Also, as gender is an important factor for adherence and successful quitting, future researches should include gender-based analysis.” (Page number - 18, Line number – 457-459)